# Holomorphic Equilibrium Propagation Computes Exact Gradients Through Finite Size Oscillations

**Axel Laborieux**[1]  **Friedemann Zenke**[1,2]

`{firstname.lastname}@fmi.ch`
[1] Friedrich Miescher Institute for Biomedical Research, Basel, Switzerland
[2] Faculty of Natural Sciences, University of Basel, Basel, Switzerland

## Abstract

Equilibrium propagation (EP) is an alternative to backpropagation (BP) that allows the training of deep neural networks with local learning rules. It thus provides a compelling framework for training neuromorphic systems and understanding learning in neurobiology. However, EP requires infinitesimal teaching signals, thereby limiting its applicability in noisy physical systems. Moreover, the algorithm requires separate temporal phases and has not been applied to large-scale problems. Here we address these issues by extending EP to holomorphic networks. We show analytically that this extension naturally leads to exact gradients even for finite-amplitude teaching signals. Importantly, the gradient can be computed as the first Fourier coefficient from finite neuronal activity oscillations in continuous time without requiring separate phases. Further, we demonstrate in numerical simulations that our approach permits robust estimation of gradients in the presence of noise and that deeper models benefit from the finite teaching signals. Finally, we establish the first benchmark for EP on the ImageNet $32 \times 32$ dataset and show that it matches the performance of an equivalent network trained with BP. Our work provides analytical insights that enable scaling EP to large-scale problems and establishes a formal framework for how oscillations could support learning in biological and neuromorphic systems.

## 1 Introduction

The backpropagation (BP) of error algorithm [1] underpins the ability of state-of-the-art deep neural networks to learn useful representations from structured data such as speech, vision, and text [2]. BP stands out as the most successful algorithm to solve the credit assignment problem in artificial neural networks [3, 4], which can be defined by the following question: How should a synaptic connection be modified in order to improve the global performance of the network to perform a task, as measured by some objective function? This is a difficult question since individual synapse may have a complicated influence on downstream processing. BP solves credit assignment through the chain rule of differentiation [1]. Although BP is efficiently implemented in software, it is difficult to conceive how BP could plausibly be implemented in biological systems. The problematic aspects are BP's use of symmetric connections and the need for two separate phases: A nonlinear forward pass that propagates neuronal activity and a linear backward pass that carries signed gradient signals [3]. These two types of processing are also inconvenient for training physical neural networks, since both should be handled by the same circuit, and explicitly propagating errors does not harness the device mismatches typical of neuromorphic hardware [5, 6]. Despite its implausibility, representations learned with BP match representations of in-vivo data [7] better than networks trained with purely biologically-motivated learning rules such as STDP [8, 9]. This discrepancy raises the question as to

36th Conference on Neural Information Processing Systems (NeurIPS 2022).

whether and how neural dynamics could implement gradient-based credit assignment, and whether it could be as effective as BP to learn useful representations [3].

Equilibrium propagation (EP) [10] is an alternative algorithm for performing credit assignment in dynamical systems that converge to a fixed point, such as energy-based models [11]. EP also proceeds in two phases: In the first phase the dynamical system is presented with static input data until the units settle into an equilibrium or fixed point. We refer to this state as the free equilibrium. In a second phase, a teaching signal slightly nudges designated output units towards a target value until the dynamics settle into a second equilibrium that is called the nudged equilibrium. EP estimates loss gradients by comparing the neuronal activity between the two equilibria. EP is appealing because the resulting learning rule is spatially local when the energy function consists of two-body interactions, as for instance in continuous Hopfield networks [11]. Furthermore EP provably approximates the true gradient in the limit of vanishing nudging [10]. More generally, the implicit differentiation carried out by EP makes it suitable for meta learning [12], where explicitly backpropagating errors through an inner optimization loop becomes prohibitive due to the high memory requirement of storing intermediate time steps for regular automatic differentiation.

Nevertheless, classic EP [10, 13, 14] has several limitations. First, EP estimates only approach the actual loss gradient in the limit of a vanishing nudging or teaching signal. This requirement makes it impractical for noisy neuromorphic systems where noise can confound small amplitude teaching signals and also unrealistic as a model for learning in the brain where feedback strongly modulates processing. Moreover, the mechanisms by which biological circuits could satisfy the requirement for separate phases remains elusive. Finally, while EP can train deep networks on CIFAR-10 [14], it has remained an open question whether it can be scaled up to larger and more complex tasks [4].

In this article, we show that by extending EP with holomorphic network dynamics it naturally estimates exact gradients for finite teaching signals. Mathematically, the exact gradients are encoded as a Fourier coefficient of adiabatic neural oscillations. This finding suggest a natural way of estimating the gradients online through suitable synaptic filtering operations which dispenses with the need for separate phases, in a similar spirit to Baldi and Pineda [15] for contrastive Hebbian learning [16]. Our main contributions are the following:

- We develop the theory of holomorphic EP (hEP) and prove that this allows computing exact gradients locally at synapses from finite teaching signal amplitudes of adiabatic oscillations.

- We numerically quantify the accuracy of our estimate and show that it outperforms classic EP, especially in the presence of substrate noise and in deep neural networks.

- We demonstrate learning with an always-on oscillating teaching signal, thereby alleviating the need for separated phases.

- Finally, we show that hEP achieves the same performance as BP in deep convolutional neural networks (CNNs) trained on CIFAR-10/100 [17], and ImageNet $32 \times 32$ [18].

## 2 Background and previous work

**Equilibrium propagation (EP).** EP [10] allows training convergent dynamical systems to optimize a loss function. We denote neuronal unit activity by the vector $\mathbf{s}$, and the learnable parameters such as weights and biases by $\boldsymbol{\theta}$ (Fig. 1a). The system's dynamics are given as the gradient of a scalar energy function $E(\boldsymbol{\theta}, \mathbf{s})$:

$$\frac{\mathrm{d}\mathbf{s}}{\mathrm{d}t} = -\frac{\partial E}{\partial \mathbf{s}}(\boldsymbol{\theta}, \mathbf{s}). \tag{1}$$

As a consequence, EP can train any energy-based models, e.g., Hopfield networks [11] to perform classification [10, 13, 14]. In classic EP, training proceeds in two phases. First, a subset of units are clamped to the input $\mathbf{x}$ and the system goes to a 'free' fixed point denoted by $\mathbf{s}_0^*$. Second, the loss function $\ell(\boldsymbol{\theta}, \mathbf{s}, \mathbf{y})$, with target $\mathbf{y}$, is scaled with a small positive nudging factor $\beta$ and added to the energy function $E$ which yields the total energy $F(\boldsymbol{\theta}, \mathbf{s}, \beta, \mathbf{y}) := E + \beta\ell(\boldsymbol{\theta}, \mathbf{s}, \mathbf{y})$. This added teaching signal causes the system to reach a second equilibrium $\mathbf{s}_\beta^*$, again by minimizing the total energy $F$. Although we write that $\ell$ takes all units $\mathbf{s}$ as argument, in practice typically only output units which encode the target label and thus serve as inputs for teaching signals are considered (Fig. 1b). The learning objective of the system is to optimize the loss function $\ell$ at the free fixed point,

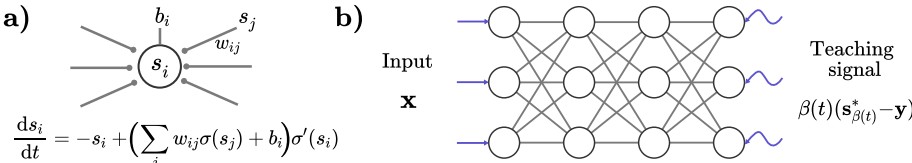

$$\frac{\mathrm{d}s_i}{\mathrm{d}t} = -s_i + \Big(\sum_j w_{ij}\sigma(s_j) + b_i\Big)\sigma'(s_i)$$

Figure 1: **a)** Schematic of the neuron model of a continuous Hopfield network [11] with holomorphic activation function $\sigma$. The neuron $s_i$ receives input both from upstream and downstream neurons $s_j$ plus a bias current $b_i$. **b)** In a typical supervised learning the input $\mathbf{x}$ is clamped, causing the network dynamics to settle into a fixed point. A complex-valued oscillating teaching signal added to the output causes neuronal activity to fluctuate around this fixed point.

which is defined by $\mathcal{L}(\boldsymbol{\theta}, \mathbf{x}, \mathbf{y}) := \ell(\boldsymbol{\theta}, \mathbf{s}_0^*, \mathbf{y})$. Scellier and Bengio [10] showed that:

$$\lim_{\beta \to 0} \frac{1}{\beta} \left( \frac{\partial F}{\partial \boldsymbol{\theta}}(\boldsymbol{\theta}, \mathbf{s}_\beta^*, \beta, \mathbf{y}) - \frac{\partial F}{\partial \boldsymbol{\theta}}(\boldsymbol{\theta}, \mathbf{s}_0^*, 0, \mathbf{y}) \right) = \frac{\mathrm{d}\mathcal{L}}{\mathrm{d}\boldsymbol{\theta}}. \tag{2}$$

This result requires $F$ to be twice continuously differentiable and assumes that one can apply the implicit function theorem to the equilibrium equation $\partial_{\mathbf{s}} F(\boldsymbol{\theta}, \mathbf{s}_0^*, 0) = 0$ (the $\mathbf{y}$ argument is hereafter omitted for clarity), so that $\beta \mapsto \mathbf{s}_\beta^*$ is a continuously differentiable map [10]. In practice, the left-hand side of Eq. (2) is estimated by finite differences [10, 13, 14, 19]. The appeal of EP for biological plausibility [3, 20] and neuromorphic hardware [15, 21–23] arises from the fact that (i) the system only needs to propagate neural activities (Fig. 1a) and (ii) in layered neural networks with a Hopfield energy [11] (Fig. 1b), the left-hand side of Eq. (2) can be computed by a Hebbian-like learning rule as the product of pre- and postsynaptic activity. In summary, EP implicitly propagates error signals through differences of neuronal activity during the two phases, whereas BP propagates error gradients explicitly [3, 20]. However, Eq. (2) only holds in the limit $\beta \to 0$ where activity differences vanish, which can pose a problem in the presence of noise or when activity differences vanish with network depth, which is related to the problem of vanishing gradients. In the following, we introduce holomorphic EP (hEP) which avoids these issues by estimating exact gradients with finite $\beta$, and thus from finite amplitude teaching signals.

## 3 Theoretical results

Our main contribution is to show that if $F$ is holomorphic, i.e., differentiable in the sense of complex variables (see Appendix A.1 for the definition), hEP computes the gradient of the objective function $\mathcal{L}$ for finite $\beta$, i.e., without requiring the vanishing nudging signals (cf. Eq. (2)). To accomplish this hEP requires a non-vanishing teaching signal that evolves 'adiabatically' in the complex plane with respect to the dynamics of the system (Fig. 1b). In other words, we require the dynamical system to relax to its equilibrium on a much shorter timescale than the timescale of the nudge.

**Derivation of holomorphic EP.** To show that hEP yields an unbiased gradient through finite adiabatic nudging, we use the same notation as in Section 2. Specifically, we extend the theory by Scellier and Bengio [10] to the complex case and to dynamical systems, or networks whose scalar function governing the dynamics is holomorphic. In line with classic EP we assume that the dynamical system has a free fixed point as described above.

**Lemma 1** (Holomorphic Equilibrium Propagation). *Let $F$ be a scalar function governing the dynamics, so that the holomorphic implicit function theorem can be applied to the fixed point equation $\partial_{\mathbf{s}} F(\boldsymbol{\theta}, \mathbf{s}_0^*, 0) = 0$, then the gradient formula of equilibrium propagation (Eq. (2)) holds in the sense of complex differentiation.*

*Proof.* The proof is an extension of the one provided by [10] for the real nudging case. The holomorphic implicit function theorem ensures that there exists an open set $U \in \mathbb{C}$ including 0 such that the implicit map $\beta \in U \mapsto \mathbf{s}_{\boldsymbol{\theta}, \beta}^*$ is holomorphic on $U$. In particular, the fixed point $\mathbf{s}_{\boldsymbol{\theta}, \beta}^*$ is defined on $U$ (see Fig. 2b for how this area looks for a toy example). The proof proceeds in two steps. First we show that the total derivatives of $F$ with respect to $\boldsymbol{\theta}$ and $\beta$ can still be interchanged for complex variables by virtue of the Schwarz theorem. Second we show that, at the fixed point, the

total derivative of $F$ with respect to $\beta$ ($\boldsymbol{\theta}$) is still equal to the partial derivatives with respect to $\beta$ ($\boldsymbol{\theta}$). To that end, we apply the chain rule of complex differentiation in:

$$\frac{\mathrm{d}F}{\mathrm{d}\beta}(\boldsymbol{\theta}, \mathbf{s}_{\boldsymbol{\theta},\beta}, \beta) = \frac{\partial F}{\partial \beta}(\boldsymbol{\theta}, \mathbf{s}_{\boldsymbol{\theta},\beta}, \beta) + \frac{\partial F}{\partial \mathbf{s}} \cdot \frac{\partial \mathbf{s}}{\partial \beta}(\boldsymbol{\theta}, \beta) + \frac{\partial F}{\partial \overline{\mathbf{s}}} \cdot \frac{\partial \overline{\mathbf{s}}}{\partial \beta}(\boldsymbol{\theta}, \beta), \tag{3}$$

where $\overline{\mathbf{s}}$ denotes the complex conjugate of $\mathbf{s}$. At equilibrium, the second term on the right hand side cancels by definition of the fixed point, and the third term is zero because $F$ is holomorphic, i.e., its derivative with respect to the conjugate variable is zero according to the Cauchy-Riemann condition [24]. The same argument holds for the derivative with respect to $\boldsymbol{\theta}$. Therefore, interchanging the total derivatives of $F$ with respect to $\beta$ and $\boldsymbol{\theta}$, and replacing the inner total derivatives by the partial derivatives, we obtain that the EP gradient formula (Eq. (2)) still holds, but for *complex* differentiation (Appendix A.1):

$$\frac{\mathrm{d}}{\mathrm{d}\beta}\bigg|_{\beta=0} \left( \frac{\partial F}{\partial \boldsymbol{\theta}}(\boldsymbol{\theta}, \mathbf{s}^*_{\boldsymbol{\theta},\beta}, \beta) \right) = \frac{\mathrm{d}}{\mathrm{d}\boldsymbol{\theta}} \frac{\partial F}{\partial \beta}(\boldsymbol{\theta}, \mathbf{s}^*_{\boldsymbol{\theta},\beta}, \beta) = \frac{\mathrm{d}\mathcal{L}}{\mathrm{d}\boldsymbol{\theta}}, \tag{4}$$

which concludes the proof (a more detailed version is in Appendix A.2). $\qquad\square$

We can now evaluate the left hand side of Eq. (4) using a Cauchy integral (Appendix A.1):

**Theorem 1** (Exact gradient from finite teaching signals)**.** *Assuming that the conditions of Lemma 1 are met and let $|\beta| > 0$ be the radius of a circular path around 0 in $\mathbb{C}$ contained in the open set $U$ on which the fixed point $\mathbf{s}^*_{\boldsymbol{\theta},\beta}$ is defined. Further assume that this path is parameterized by $t \in [0, T] \mapsto \beta(t) = |\beta|e^{2\mathrm{i}\pi t/T}$, where $\mathrm{i}$ is the imaginary unit. Then the loss gradient is given by:*

$$\frac{\mathrm{d}\mathcal{L}}{\mathrm{d}\boldsymbol{\theta}} = \frac{1}{T|\beta|} \int_0^T \frac{\partial F}{\partial \boldsymbol{\theta}} \left( \boldsymbol{\theta}, \mathbf{s}^*_{\boldsymbol{\theta},\beta(t)}, \beta(t) \right) e^{-2\mathrm{i}\pi t/T} \mathrm{d}t. \tag{5}$$

The full proof is given in Appendix A.3. Theorem 1 guarantees that given holomorphic dynamics we can dispense with the requirement of vanishing teaching signal $|\beta| \to 0$ in the limit of 'adiabatic' nudging which corresponds to integrating over infinitely many fixed points with a *finite* $|\beta|$. Note, that complex-valued teaching signals $\beta \in \gamma$ produce fixed points in the complex plane computed through the same equations as in the real case (see Appendix B). In particular, multi-layered neural networks (Fig. 1b) can be trained by using the continuous Hopfield dynamics [11]. The trainable parameters are the weights and biases $\boldsymbol{\theta} = (w_{ij}, b_i)$ and the total energy function $F$ is given by:

$$F(\boldsymbol{\theta}, \mathbf{s}, \beta, \mathbf{y}) = \frac{1}{2} \sum_i s_i^2 - \frac{1}{2} \sum_{i \neq j} w_{i,j} \sigma(s_i)\sigma(s_j) - \sum_i b_i \sigma(s_i) + \beta \ell(\boldsymbol{\theta}, \mathbf{s}, \mathbf{y}). \tag{6}$$

If the activation function $\sigma$ is holomorphic, which is true in the case of sigmoid functions, the same $F$ (Eq. (6)) can be evaluated with complex $\beta$, and we can apply Eq. (5) to obtain:

$$-\frac{\mathrm{d}\mathcal{L}}{\mathrm{d}w_{ij}} = \frac{1}{T|\beta|} \int_0^T \sigma_i^*(t)\sigma_j^*(t) e^{-2\mathrm{i}\pi t/T} \mathrm{d}t, \tag{7}$$

where $\sigma_i^*(t) := \sigma(s^*_{i,\beta(t)})$. Therefore, assuming a $T$-periodic teaching signal (Fig. 1b), the gradient is proportional to the first exponential Fourier coefficient of the product of the oscillating activities. Although this formulation assumes complex neuronal output, we show in Appendix A.4 that the gradient in Eq. (7) can be expressed in terms of the real part or imaginary part only. The complex teaching signal is therefore best thought of as a way to produce unbiased neuronal oscillations on the nudging timescale. In the next section, we numerically estimate the Fourier coefficient of Eq. (5) with a fixed number of points $N$ on the circle which we use to train networks in the subsequent experiments and to compare it to the actual loss gradient computed with automatic differentiation.

**Numerical estimation of the loss gradient as a Fourier coefficient.** Next, we explain how to estimate the gradient from the corresponding Fourier coefficient (Eq. (5)). In practice, we use a Riemann sum to compute the integral numerically. We fix the nudging radius $|\beta| > 0$ such that the circular path lies in the domain $U$ in which equilibria exist as described in Theorem 1. We sample the path with $N \geq 2$ nudging points $\{\beta_k := |\beta|e^{2\mathrm{i}\pi k/N} \; ; \; k \in [0, ..., N-1]\}$, and define the estimator:

$$\hat{\nabla}(N) := \frac{1}{N|\beta|} \sum_{k=0}^{N-1} \frac{\partial F}{\partial \boldsymbol{\theta}} \left( \boldsymbol{\theta}, \mathbf{s}^*_{\beta_k}, \beta_k \right) e^{-2\mathrm{i}\pi k/N}. \tag{8}$$

We have that $\hat{\nabla}(N) \underset{N\to\infty}{\longrightarrow} \frac{d\mathcal{L}}{d\boldsymbol{\theta}}$, and the remaining bias term when using $N$ points is:

$$\hat{\nabla}(N) - \frac{d\mathcal{L}}{d\boldsymbol{\theta}} = \sum_{p\equiv 0\ (N)}^{\infty} \frac{C_{p+1}|\beta|^p}{(p+1)!}, \tag{9}$$

where $C_p$ is the $p$-th derivative in 0 of the function $\beta \mapsto \partial_{\boldsymbol{\theta}} F(\boldsymbol{\theta}, \mathbf{s}^*_\beta, \beta)$ (see Appendix A.5 for the proof). The bias term in Eq. 9 converges to zero with increasing $N$ because it is a sub-sum of the $(N+1)$-th order remainder of the series expansion of $\beta \mapsto \partial_{\boldsymbol{\theta}} F(\boldsymbol{\theta}, \mathbf{s}^*_\beta, \beta)$ in 0. The rate of convergence depends on the $C_p$ coefficients and the radius $|\beta|$. In the case $N = 2$, the estimate of Eq. (8) coincides with the 'symmetric' estimate of Laborieux et al. [14]. However, the bias term on the right hand side of Eq. (9) is only valid when the dynamics are holomorphic. Next, we illustrate the approach in a toy experiment, and list three practical improvements brought by hEP.

## 4 Experiments

In all the experiments, we used the discrete setting of convergent recurrent neural networks of [13], and the readout scheme of [14] for optimizing the cross-entropy loss function with EP (Appendix B). All simulations were implemented in Jax [25] and Haiku [26] (Apache License 2.0). The datasets were obtained from the Tensorflow datasets library [27]. Our code is publicly available on GitHub[1]. The details of simulations and hyperparameters can be found in Appendix E and F.

**Demonstration of holomorphic Equilibrium Propagation on a single data point.** To provide the first numerical validation of Theorem 1 while also allowing us to gain intuitions about dynamics of individual neurons, we implemented a small-scale multi-layer perceptron (MLP) with layer dimensions 6-4-4-4, including input and output layers. The activation function was a shifted sigmoid, which is holomorphic (Appendix B). The network was fed with a single datapoint, namely a randomly sampled point from a Gaussian and a random one-hot target. The dark blue region in Figure 2b shows the map of complex $\beta$ for which the network settles to a fixed point after 200 time steps. We found experimentally that the area in the complex plane where stable fixed points exist strongly depends on the activation function and the weight initialization (see Appendix D). As $\beta$ evolves on the circle of radius 0.1 ($N = 24$) hidden layer neurons settle into different equilibrium points in the complex plane (Fig. 2a). We observed that while the teaching signal $\beta(t) = |\beta|e^{2i\pi t/T}$ was purely sinusoidal, the non-linearity of the network induces neural oscillations that are not purely sinusoidal. Nevertheless, the gradient is contained in the first mode of these non-linear oscillations (Fig. 2c).

To understand how the gradient is computed accurately when the magnitude of the teaching signal is increased, we recorded the adiabatic product of activities $\sigma_i^*(t)\sigma_j^*(t)$ for one pair of neurons (dark blue) over one teaching period and for increasing values of $|\beta|$ (Fig. 2c). In the case $|\beta| = 0.001$, the perturbation induced by the sinusoidal teaching signal is also purely sinusoidal, and the gradient magnitude is simply the radius of the circle. However, when $|\beta|$ is increased to 0.01, the linear approximation of the perturbation becomes less accurate, because higher powers of $\beta$ become significant in the series expansion around the free fixed point. The gradient could still be well approximated by taking the mean of the two radii corresponding to real positive and negative $\beta$, as done by Laborieux et al. [14]. Increasing $|\beta|$ further to 0.1 and 0.5 yields an even more deformed perturbation, but the gradient is still correctly contained in the first Fourier coefficient of the perturbation. hEP breaks down when $\beta$ reaches amplitudes for which the corresponding path intersects with areas in which no stable equilibrium exists (cf. Fig. 2b, light areas, and Fig. 2d). However, as we will see in the next sections, the finite teaching amplitudes are beneficial when the neuronal dynamics are subject to noise and when training deep neural networks.

**Holomorphic EP can estimate the gradient in continuous time.** Our theoretical findings allow us to revisit in a principled way an idea introduced by Baldi and Pineda [15] for a continuous-time-implementation of contrastive Hebbian learning [16] and learning rules looking to maximize slowness [28–30]. In this context, the oscillating teaching signal is always-on, and the dynamical network is governed by three mechanisms acting on distinct timescales. The smallest timescale $T_{\text{dyn}}$ is the typical time needed by the network to reach its fixed point. The second timescale is the period of

---

[1] https://github.com/Laborieux-Axel/holomorphic_eqprop

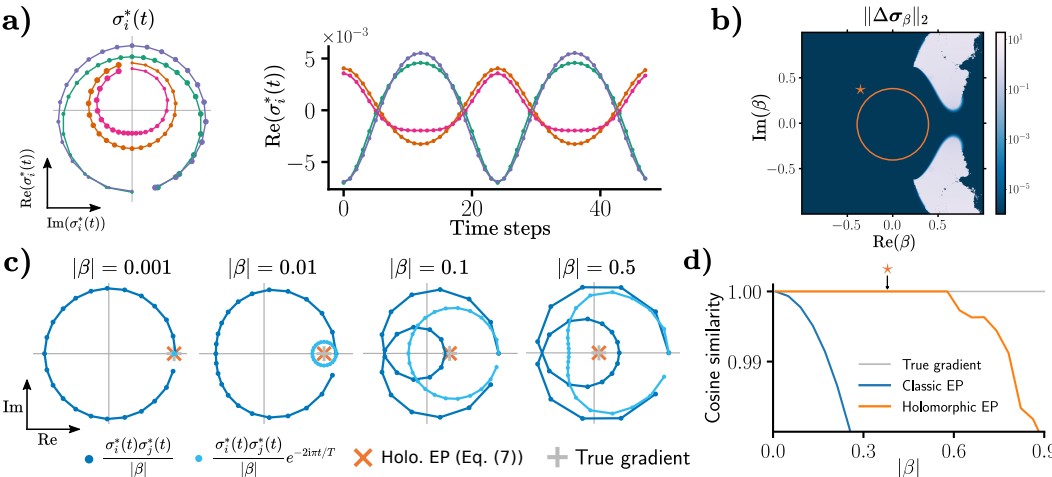

Figure 2: Overview of hEP for a small MLP. **a)** Neural oscillations sampled at $N = 24$ points in the complex plane relative to the free fixed-point in the center (left, point sizes increasing with time). The corresponding real part values over two periods (right). **b)** Map of the euclidean norm between two consecutive steps of the dynamics in the complex plane spanned by $\beta$. The map describes the network's dynamical stability. Dark blue corresponds to regions with stable fixed points, while light blue indicates lack of stability. **c)** Adiabatic correlations between two neurons for different $|\beta|$ values (dark blue dots). Filtering the mode over the period $T$ of the teaching oscillation gives the light blue dots, the temporal mean of which (orange cross) is the gradient (grey plus sign). **d)** Cosine similarity with the true gradient of hEP (orange), and classic EP [10] (blue). The $\star$ marks the teaching radius $|\beta|$ of the path in panel b). hEP breaks down when the path passes through unstable (light) regions.

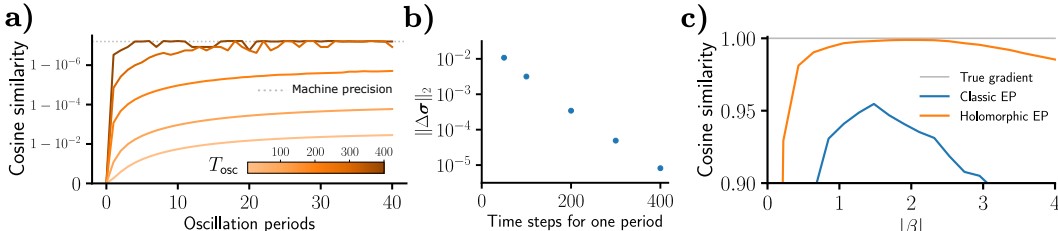

Figure 3: **a)** Cosine similarity between the true gradient obtained with BP through time and the online estimate ($N = 10$) as a function of oscillation periods. Different curves correspond to different oscillation periods (darker color indicates larger $T_{\text{osc}}$). $T_{\text{plas}} \approx 10 T_{\text{osc}}$ is enough to accurately estimate the gradient. **b)** Measure of the residual convergence of the network as a function of the oscillation period $T_{\text{osc}}$, showing that $T_{\text{dyn}} \approx 400/10 = 40$ time steps. **c)** Cosine similarity of hEP ($N = 15$) and classic EP with the true gradient as a function of $|\beta|$ with neuronal output noise. All panels used the same MLP with two hidden layers of 256 neurons each, fed with a minibatch of ten MNIST samples.

one teaching oscillation $T_{\text{osc}}$, and the third timescale is the number of periods after which synaptic plasticity occurs $T_{\text{plas}}$. The gradient can be estimated online by:

$$\widetilde{\nabla}(T_{\text{plas}}) := -\frac{1}{T_{\text{plas}}|\beta|} \int_0^{T_{\text{plas}}} \sigma_i(t)\sigma_j(t)e^{-2i\pi t/T_{\text{osc}}}\mathrm{d}t, \qquad (10)$$

which converges to the gradient if $T_{\text{dyn}} \ll T_{\text{osc}} \ll T_{\text{plas}}$ (see Appendix A.6). This expression is similar to Eq. (5.2) in [15]. However their teaching signal oscillates discretely between 0 and 1, and therefore produces a biased estimate of the gradient. To test the influence of the oscillation timescale $T_{\text{osc}}$ on the online estimate of Eq. (10), we compared the online estimation of the gradient over several periods between several values of $T_{\text{osc}}$. To this end, we used a MLP with two hidden layers with 256 units each, which we fed with a minibatch of MNIST data [31]. We observed that the gradient could be accurately estimated in a few periods for high enough $T_{\text{osc}}$ (Fig. 3a, dark curves). However, when the oscillations were too fast, a non-vanishing bias remained in the gradient estimates even for many

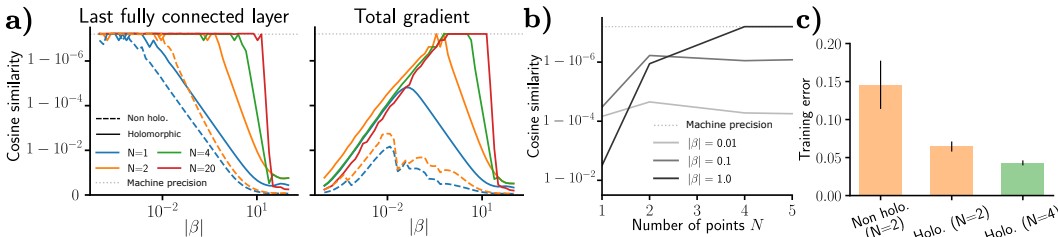

Figure 4: **a)** Cosine similarity between hEP and the true gradient in a holomorphic (plain lines) seven-layer VGG-like CNN [32], and a non-holomorphic version using max pooling and ReLUs (dashed lines). Input data is a minibatch of ImageNet $32 \times 32$ [18] consisting of 10 images. The left plot shows a comparison of the gradient with respect to the parameters of network's output layer. The right plot takes into account the gradient with respect to all network parameters. **b)** Cosine similarity in function of $N$ for three teaching amplitude values. Increasing $N$ is only required for higher amplitudes. **c)** Average training error on CIFAR-10. The average is calculated over three random initializations and error bars correspond to two standard deviations.

periods (Fig. 3a, lighter curves). This bias is in all likelihood due to the inability of the system to reach the fixed point (Fig. 3b). Finally, we found that given appropriate period timings, hEP used in the online setting can train a network on MNIST (Table 1). Importantly, the online formulation of hEP allows to dispense with the requirement of strictly separate learning phases by replacing them with separate plasticity mechanisms acting on different timescales.

Table 1: MNIST validation errors in % for classic EP [10], hEP, and online hEP, with and without noise. Results are averages ($n = 3$) $\pm$ stddev. For training errors see Table 4 in Appendix E.

| Noise | Class. EP, $|\beta| = 0.1$ | Class. EP, $|\beta| = 0.4$ | hEP, $|\beta| = 0.4$ | Online hEP |
|---|---|---|---|---|
| Noise-free | $1.87 \pm 0.01$ | $2.24 \pm 0.05$ | $1.97 \pm 0.08$ | $2.05 \pm 0.02$ |
| With noise | $88.7 \pm 0.0$ | $3.01 \pm 0.1$ | $1.96 \pm 0.07$ | $1.91 \pm 0.16$ |

**Finite size teaching oscillations provide robustness to noise.** To analyze hEP's robustness to noise, we injected a small-amplitude zero-mean Gaussian noise to each neuron in the network in addition to the input from other neurons. We then used a single minibatch from the MNIST dataset to compute gradient estimates using classic EP and hEP. The latter was computed by using one realisation using $N = 15$ points, whereas the classic EP estimate was computed using the free and nudged fixed points each averaged over $\lceil N/2 \rceil$ to provide a fair comparison. We found that for small $\beta$ when noise amplitudes were comparable to the activity changes caused by teaching oscillations the gradient estimate diverged from the true gradient of the noise-free system (Fig. 3c). To some extent this effect could be mitigated by choosing a finite teaching signal ($\beta \gg 0$) [12]. However, since $\beta$ also increases the bias for classic EP this creates a trade-off between choosing $\beta$ either too small or too large. Importantly, even for the optimal choice of $\beta$, classic EP did not accurately approximate the gradient of the noise-free system. In contrast, hEP thanks to its robustness to finite teaching signals did provide an accurate estimate of the gradient despite the noise. We verified that hEP is indeed more robust to noise than classic EP when used to train the network (Table 1). Thus, hEP combined with finite teaching amplitudes provides an effective way for training noisy computational substrates.

**Holomorphic EP matches BP performance on large-scale vision benchmarks.** To test hEP's ability to train deep neural networks, we first investigated the influence of the number of fixed points $N$ and the teaching amplitude $|\beta|$ on the approximating quality $\hat{\nabla}(N)$ of the loss gradient in a seven-layer VGG-like architecture ([32]; Fig. 4a,b). We ensured holomorphic dynamics by using softmax pooling layers [23] instead of the non-holomorphic max pooling, and by relying on sigmoid weighted linear units (dSiLU) [33] (see Appendix B) instead of standard ReLUs. We first considered a minibatch of ten images from the ImageNet $32 \times 32$ dataset [18] and computed the gradient using hEP as well as BPTT for reference. Here, our estimate (Eq. (8)) was computed by first letting the network settle to the free fixed ($\beta = 0$), and then running the phases with complex $\beta$. We found that the change to holomorphic dynamics already improved upon the gradient estimates used in previous

work [14, 19]. Moreover, we observed that for the last layer increasing $N$ only extended the range of usable teaching magnitudes $|\beta|$, but did not improve the quality of the gradient estimate. This phenomenon can be understood from Eq. (9), since higher $N$ reduces the bias term considerably, which accommodates higher teaching magnitude $|\beta|$. However, larger amplitudes tended to improve the total gradient, particularly in deep layers where small teaching magnitudes were not enough to produce sufficient error signals (see Appendix C for details). Additionally, larger $N$ were only required when using a higher teaching amplitude (Fig. 4b). Finally, we tested how the gradient quality impacts the network training accuracy on CIFAR-10. We observed that the non-holomorphic VGG was unable to reach low training error (Fig. 4c), which is consistent with the poor gradient quality (Fig 4a). Changing to a holomorphic architecture with the same number of points resulted in a substantial improvement of training accuracy, which was further boosted when training with $N = 4$ consistent with our theory.

Table 2: Validation accuracy of BP and hEP. All values are averages $(n = 3) \pm$ stddev.

|  | CIFAR-10 | CIFAR-100 | | ImageNet $32 \times 32$ | |
|---|---|---|---|---|---|
|  | Top-1 (%) | Top-1 (%) | Top-5 (%) | Top-1 (%) | Top-5 (%) |
| BP | $88.3 \pm 0.1$ | $62.0 \pm 0.5$ | $86.2 \pm 0.1$ | $37.2 \pm 0.4$ | $60.9 \pm 0.1$ |
| hEP | $88.6 \pm 0.2$ | $61.6 \pm 0.1$ | $86.0 \pm 0.1$ | $36.5 \pm 0.3$ | $60.8 \pm 0.4$ |

Finally, we wondered whether hEP could train deep neural networks on large-scale datasets, which has remained an open problem for most if not all alternative algorithms to BP [4]. To this end, we trained a five-layer CNN based on the VGG architecture [32] on multiple vision benchmarks including CIFAR-10, CIFAR-100 [17], and the $32 \times 32$ pixel version of ImageNet [18], which contains 1.2 million data points and 1000 classes like the full ILSVRC dataset [34]. In all cases we found that the validation accuracy reached by networks trained with BP and hEP using only two fixed points ($N = 2$) were identical within their uncertainties (Table 2). Note that the networks trained with BP were the feed-forward equivalent of the holomorphic networks used with EP, but with ReLUs instead of dSiLU, which did not give satisfactory results. Thus, hEP permits training deep CNNs on ImageNet $32 \times 32$ to comparable performance levels as standard BP.

## 5   Discussion

We have introduced hEP which extends classic EP by computing exact loss gradients through integration over finite size adiabatic neuronal oscillations caused by a teaching signal (Section 3). Importantly, such integration can be accomplished online with purely local learning rules which makes it an exciting theoretical framework for studying learning in the brain where oscillations are ubiquitously observed [35, 36]. In practice we found that numerically evaluating a small number of points during one oscillation cycle provides an excellent gradient approximation that outperforms classic EP and thanks to the finite oscillation amplitude is robust to noise, which is an advantage for training neuromorphic hardware systems [21, 22, 37]. Additionally, the possibility of using finite teaching signals is conducive for training deep CNNs, where infinitesimal teaching signals as used by classic EP, may vanish (Section 4).

A body of previous work has attempted to reconcile BP with neurobiology [3, 38]. EP is most closely related to classic theories of predictive coding (PC) [39–45] which similarly assumes convergent network dynamics cast into an energy minimization problem. PC further assumes that errors are encoded in neuronal dynamics, dedicated dendritic compartments, or separate temporal phases [20, 40, 46, 47]. In a similar vein, Target Propagation (TP) [48–50] assumes locally encoded error signals which in some cases are obtained by iterating approximate inverses, a property reminiscent of EP [51–54] which comes with theoretical guarantees [55]. However, all previous EP studies and most of the works above, with two notable exceptions [38, 50], were all limited to small-scale problems [4]. In contrast, we demonstrate in this article that hEP scales to ImageNet $32 \times 32$.

While the ability to run hEP online makes it an appealing model for credit assignment in biological neural networks, this interpretation has several notable shortcomings. First, hEP requires complex-valued neuronal outputs and a holomorphic dynamical system which precludes the use of max pooling and ReLUs and hampers a direct comparison to neurobiology. However, we found that holomorphic

alternatives exists which empirically yield comparable performance. Moreover, complex outputs have a long-standing tradition in computational neuroscience where they appear in variations of Hopfield networks [56–58], in the framework of theta neurons [59], and phasor networks [60] where they are used to describe oscillatory neuronal dynamics. It is possible to interpret hEP within such frameworks. For instance, it is straightforward to interpret complex neuronal output as oscillating activity with a defined amplitude and relative phase to some reference signal accessible to the entire neuronal population. Such a signal could be provided by neuromodulators such as acetylcholine which has been implicated in neural oscillations [61]. Within our framework, the oscillatory teaching signal then corresponds to a slow phase precession between the neuronal activity and the reference. Importantly, such a mechanism implies a hierarchy of oscillation frequencies. Such different oscillations are known to exist in the brain, e.g., theta (4–8 Hz) and gamma (30–70 Hz), but their precise purpose remains elusive. While establishing formal circuit-level equivalences with hEP will require future work, the principled link between oscillatory activity, learning and memory as developed in this article seems promising. Like other algorithms hEP requires symmetric synaptic connectivity between layers which seems biologically implausible. While theoretical guarantees for exact gradient computation are lost without strict symmetry [62], it may not be required for learning [63–65]. Alternatively, symmetry may be acquired through plastic feedback connections [66, 67]. Although our approach neither uses time-varying input nor neuronal spiking dynamics, spiking extensions to EP have been proposed [68, 69]. However, applying our theory to time-varying tasks requiring memory will require additional architectural modifications and theoretical concepts [70] and establishing links to present spike-based approaches [38, 71–73].

Our work augments classic EP with desirable properties for potential neuromorphic applications, which promise power-efficient and equitable artificial intelligence (AI) at the edge and in IoT devices [74–77]. While current software implementation of EP are generally slow compared to backprop, its appeal lies in its potential for training physical networks on future neuromorphic mixed-signal devices that are incompatible with backprop, but achieve settling times on the order of nanoseconds [21, 37]. As exciting as such developments are, they also risk negative societal impacts, e.g., through mass surveillance or allowing AI systems with potentially discriminatory biases to permeate our everyday lives further. A transparent research strategy and taking into account ethical considerations early during product design will be essential to avoid such adverse outcomes. On the upside, our theoretical work further consolidates EP as a conceptual framework for understanding the brain, a fundamental requirement to inform future biomedical research targeted at nervous system disorders.

## Acknowledgements

We thank all members of the Zenke Group for comments and discussions. We also thank the anonymous reviewers for their useful feedback. This project was supported by the Swiss National Science Foundation [grant number PCEFP3_202981] and the Novartis Research Foundation.

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
