# OpenReview forum: "Holomorphic Equilibrium Propagation Computes Exact Gradients Through Finite Size Oscillations"
_NeurIPS.cc/2022/Conference — NeurIPS 2022 Accept_

### Official Review · Reviewer_mSnz · 2022-07-01

**Rating:** 8
**Confidence:** 4
**Soundness:** 4 excellent
**Presentation:** 4 excellent
**Contribution:** 4 excellent

**Summary:**

The authors begin by showing that the classic equilibrium propagation can be extended to the case where the network layers are holomorphic dynamical systems. The essential result which follows is that the gradient with respect to parameters can be transformed to a contour integral on a circle around zero, which is equivalent to the first Fourier coefficient of the derivative of the nudged energy function with respect to network parameters. The central claim is that this equivalence eliminates the requirement that the "nudge size" go to zero. The authors define a straightforward numerical approximation to the gradients (equivalent to discrete Fourier transform), and show that the bias of this approximation goes to 0 as the sample rate of the Fourier transform goes to infinity.

The authors' toy experiment shows that holomorphic EP is capable of learning and that the cosine similarity of the approximated gradients is significantly higher than with classic EP.

The authors' theoretical investigations also reveal that holomorphic EP can be adapted to an online learning algorithm, assuming that the time scale of the layer dynamics is much less than the time scale of the oscillations computing the Fourier transform, which is in turn much less than the time scale of the weight updates. The authors validate this result with the MNIST task and show that the online approximation of the gradients are still significantly more accurate than the classic EP algorithm. They also compare classic, homlomorphic, and online homlomorphic EP on the MNIST task and claim that holomorphic EP has greater robustness to noise.

Finally, the authors validate their algorithm by adapting a CNN architecture to be holomorphic and showing that the holomorphic EP training yields virtually the same results as BP training on a standard CNN.

**Questions:**

A major numerical drawback of holomorphic functions is the presence of poles and unbounded behavior. Both the sigmoid and dSilU activation functions will have poles where ever the exponential function takes the value -1. Why do these poles not cause any kind of instability in the numerical implementation?

Since the learning algorithm can be adapted to be online, it was a bit of a disappointment to not see any time-domain tasks. The authors mention this as a limitation in their discussion. However, why would the algorithm still not work if the time scale of the task was much slower than the time scale $T_{dyn}$ of the layer dynamics? Was this ever tested?



**Limitations:**

The authors adequately addressed potential societal impact and some of the limitations of their work. However, only in the final appendix did I find a note that holomorphic EP was *much* slower at learning than BP. Especially since the authors mention edge and IoT devices in their discussion, I think a discussion of computational cost should also be moved to the discussion. Just how much longer should we expect holomorphic EP to take to train and evaluate? Was it only for ImageNet, and why would that be the case? It was not made clear whether or not 64 bits was required for each real and imaginary part or for the entire complex number, this is an important factor which would certainly affect performance.

**Strengths And Weaknesses:**

The paper presents a very novel idea which greatly increases the power of a local learning rule using the elegance of complex analysis.

The proof of theorem 1 in the main text could be presented more clearly. It seemed as though the authors were implying that the logic of the complex chain rule needed to be applied to $\theta$ as well as $\beta$. The proof in the appendix is much more clear, I think that one thing the authors could do to improve presentation in the main text would be to denote $s_{\theta,\beta}$ instead of $s_{\beta}$.

The derivation of the weight gradients presented was only with respect to a mean-squared error loss. However, for the computer vision tasks I would assume cross entropy loss would be required. Was an equivalent version of Eq. 8 derived for cross entropy anywhere? Eq. 8 seemed out of place since the main tasks are classification. If I understood correctly, Eq. 8 is the basis for the derivation of the online learning algorithm, is the use of MSE in the online learning essential or inconsequential? It would be strange if the online training of MNIST used MSE. Was the JAX autodiff engine used to compute the gradient of the nudged energy function with respect to parameters, in the case where MSE was not used?

The toy task was not adequately explained in the main text, what exactly are the inputs and targets? The way it sounded was that the network is given noise as input and is supposed to output a single value. This is very trivial, even for a toy task. Were there multiple Gaussians? Was the task to classify which samples came from which distribution? How many classes were there?

Another minor clarity issue is the labelling of Figures 2d and 3c versus 4a and S6. Why did the authors choose to present 1 - cos similarity instead of just the similarity itself? This made the plots look very surprising at first.

Overall, this is a very interesting paper which I would very strongly accept on the condition that its clarity issues are fixed.

---

> ### Author Response · Authors · 2022-08-02
> **Response to reviewer mSnz Part 1**
>
> Dear reviewer mSnz,
>
> We would like to thank you for your positive comments and detailed feedback. We describe below the proposed modifications to the paper in order to address all the points you raised. We believe the paper gained in clarity thanks to this feedback. **All line, equation, and table numbers are given with respect to the revised version.**
>
> > The proof of theorem 1 in the main text could be presented more clearly. It seemed as though the authors were implying that the logic of the complex chain rule needed to be applied to theta as well as beta. The proof in the appendix is much more clear, I think that one thing the authors could do to improve presentation in the main text would be to denote $s_{\theta,\beta}$ instead of $s_{\beta}$.
>
> Thanks for raising this point. We modified the proof in the main text to make it more similar to the full proof of the appendix. In particular we now write $s_{\theta,\beta}$ instead of $s_{\beta}$ as suggested.
>
> > The derivation of the weight gradients presented was only with respect to a mean-squared error loss. However, for the computer vision tasks I would assume cross entropy loss would be required. Was an equivalent version of Eq. 8 derived for cross entropy anywhere? Eq. 8 seemed out of place since the main tasks are classification. If I understood correctly, Eq. 8 is the basis for the derivation of the online learning algorithm, is the use of MSE in the online learning essential or inconsequential? It would be strange if the online training of MNIST used MSE. Was the JAX autodiff engine used to compute the gradient of the nudged energy function with respect to parameters, in the case where MSE was not used?
>
> Thanks for asking this question. Equation (7) is the learning rule and it does not depend on which loss function is used at the network’s output, because the partial derivative of $F$ with respect to the weights remains equal to the product of neural activities. What does change is the loss term multiplied by $\beta$ in the energy Eq. (6) and therefore the equilibrium states of the network. We actually use cross entropy loss for all experiments including online learning. The occurrence of MSE in Eq. (7) in the old revision was only used to explain the theory intuitively and compactly. We now realize that this was misleading and we adapted what is now Eq. (6) accordingly. Further, we show the full energy including cross entropy loss in Appendix B where we previously only showed the resulting dynamics in discrete time.
>
> > The toy task was not adequately explained in the main text, what exactly are the inputs and targets? The way it sounded was that the network is given noise as input and is supposed to output a single value. This is very trivial, even for a toy task. Were there multiple Gaussians? Was the task to classify which samples came from which distribution? How many classes were there?
>
> Our apologies that this was not clear. The term “toy experiment” might have been a misnomer. We literally use a single datapoint and target here. This is done mainly for illustration purposes as it allows us to plot all neurons for all inputs in Figure 2, which would be impossible for any real task. In some way it is the EP equivalent of the old machine learning debugging trick of overfitting on a single minibatch, where in our the batch size one. Importantly, there is no learning in this toy experiment and weights are not updated and we only use it to check that the hEP gradient for finite $\beta$ aligns with the BPTT gradient. Of course it would be easy to come up with a more sophisticated task. However, we prefer to stick with this simplistic example for didactic and illustration purposes. That these findings generalize to real-world tasks is shown in subsequent figures on MNIST and ImageNet32. To avoid any confusion, we updated the description of the “toy experiment” section to clearly state our rationale.
>
> Specifically, we now write:
>
> *“To provide the first numerical validation of Theorem 1 while also allowing us to gain intuitions about dynamics of individual neurons, we implemented a small-scale multi-layer perceptron with layer dimensions 6-4-4-4, including input and output layers. The activation function was a shifted sigmoid, which is holomorphic (see Appendix B). The network was fed with a single datapoint, namely a randomly sampled point from a Gaussian and a random one-hot target”*
>
> > Another minor clarity issue is the labelling of Figures 2d and 3c versus 4a and S6. Why did the authors choose to present 1 - cos similarity instead of just the similarity itself? This made the plots look very surprising at first.
>
> Thanks for raising this. We now consistently plot the cosine similarity throughout the paper.

---

> ### Author Response · Authors · 2022-08-02
> **Response to reviewer mSnz Part 2**
>
> > A major numerical drawback of holomorphic functions is the presence of poles and unbounded behavior. Both the sigmoid and dSilU activation functions will have poles where ever the exponential function takes the value -1. Why do these poles not cause any kind of instability in the numerical implementation?
>
> Thanks for asking this question. Indeed, the holomorphic activation functions we used have poles and, indeed, these poles can cause instability. This can be seen, for instance, in the divergent areas (light color) in Figs 2.b) and Fig 7b-f). Although we did not systematically study this phenomenon, we think that these areas of instability are at least in part due to the poles of the activation function. However, we found, in practice, that for reasonable choices of activation functions, the temperature parameter of the Softmax pooling, and weight initialization the networks we studied showed large “stable areas” around the free fixed point, enough to compute the gradient with finite $\beta$. We think it is interesting, and deserves future study, something we plan on doing. For the sake of this article, we now discuss this topic in the revised Appendix D.
>
> *“These diverging areas could be due to the poles of the activation functions used. For example, the sigmoid function $z \mapsto 1/(1+e^{-z})$ has $\{(2k+1)\mathrm{i}\pi ; k \in \mathbb{Z}\}$ as a set of poles where it diverges. Although we did not systematically study this phenomenon in this work, we strongly suspect that these unstable areas are partly the result of the teaching signal being too strong or the weights being poorly distributed, thereby driving the complex neural activities near to the poles. To some extent, the poles can be brought farther by introducing a coefficient in the exponential, but it results in flatter activation functions on the real axis, so a trade-off should be found. In practice, we found that choosing reasonably the activation function, weight initialization, and teaching radius $|\beta|$ lead to enough stable areas around 0 to compute the gradient.”*
>
> > Since the learning algorithm can be adapted to be online, it was a bit of a disappointment to not see any time-domain tasks. The authors mention this as a limitation in their discussion. However, why would the algorithm still not work if the time scale of the task was much slower than the time scale $T_{\text{dyn}}$ of the layer dynamics? Was this ever tested?
>
> We agree, if the network is at quasi equilibrium throughout an entire input-output trajectory mapping there should be no problem, although we did not explicitly try this. But it would be easy to do. However, and that’s what we discuss as a limitation in the discussion, if the network has to self-generate a trajectory from an initial cue, or is not allowed to settle into equilibrium, then we expect hEP to break down. Some ideas have, for instance, been studied for Hopfield networks, where they require fast and slow weights or neuronal adaptation variables to provide a temporal context and store sequences of patterns. We now edited the discussion to make this distinction more clear. Specifically, we write:
>
> *“However, applying our theory to time-varying tasks requiring memory will require additional architectural modifications and theoretical concepts [69] and establishing links to present spike-based approaches [38, 70–72].”*

---

> ### Author Response · Authors · 2022-08-02
> **Response to reviewer mSnz Part 3**
>
> > The authors adequately addressed potential societal impact and some of the limitations of their work. However, only in the final appendix did I find a note that holomorphic EP was much slower at learning than BP. Especially since the authors mention edge and IoT devices in their discussion, I think a discussion of computational cost should also be moved to the discussion. Just how much longer should we expect holomorphic EP to take to train and evaluate? Was it only for ImageNet, and why would that be the case?
>
> Yes, for current software frameworks EP is generally slower than BP. However, this discrepancy is due to the different underlying network architectures. While BP trains conventional ANNs, EP trains a recurrent version of a related network that needs to settle to equilibrium which requires the bulk of simulation time. While this time could be reduced in software by using smarter ways to reach the fixed point such as approximated second order methods (pseudo Newton) or others such as Anderson acceleration (see [Bai et al. NeurIPS 2019]), the real goal of EP is to make its mark in novel compute devices beyond von Neumann computers, that seek to emulate the dynamical system [Foroushani et al, ISCAS 2020] rather than simulating it in software. On such physical on-device network implementations, the use of backprop is often impossible due to physical constraints, whereas the convergence speed to a fixed point can be dramatically faster, potentially down to the nanosecond scale [Kendall et al, arXiv:2006.01981]. As suggested, we added a sentence in the discussion to mention these aspects.
>
> Specifically, we write:
>
> *“While current software implementation of EP are generally slow compared to backprop, its appeal lies in its potential for training physical networks on future neuromorphic mixed-signal devices that are incompatible with backprop, but achieve settling times on the order of nanoseconds [21, 37].”*
>
> > It was not made clear whether or not 64 bits was required for each real and imaginary part or for the entire complex number, this is an important factor which would certainly affect performance.
>
> Our current Jax implementation requires 64 bits for the entire complex number, that is 32 bits for the real part and 32 bits for the imaginary part. We make it explicit in the revised appendix.

---

> > ### Comment · Reviewer_mSnz · 2022-08-03
> > **Response to rebuttal**
> >
> > Thank you for the very comprehensive responses to my questions and comments. Since you fixed many things I found unclear, I increased the presentation rating from 3 to 4.
> >
> > I am personally very enthusiastic about the results presented here, and I find the potential relationship to hippocampal learning very compelling. The fact that gradients with cosine similarity 1 to backprop can be achieved without actually using backprop certainly has a high impact on multiple fields of machine learning. However, since the algorithm is strictly slower than backprop on GPUs, and is intended for unconventional hardware, I would say the paper's *excellent* impact is limited to the fields of neuromorphic computing / bioplausible learning. Therefore, I will retain my overall review of 8.

---

### Official Review · Reviewer_EuCr · 2022-07-07

**Rating:** 8
**Confidence:** 3
**Soundness:** 4 excellent
**Presentation:** 4 excellent
**Contribution:** 3 good

**Summary:**

The authors present `holomorphic equilibrium propagation' (holomorphic EP), an extension of equilibrium propagation for networks with holomorphic activation functions, which estimates the gradient of the loss using an approximation of the complex Cauchy integral. This avoids the need for an infinitesimal teaching signal, making the method more robust to noise.

**Questions:**

For someone who is not an expert, Figure 2 was challenging for me to interpret. Perhaps the figures can be connected more directly to the equations?

Minor questions/comments:

Line 143: it's not clear to me why $\text{d}\beta=(2\text{i}\pi\beta/T)\text{d}t$. Is there a $e^{2\text{i}\pi t/T}$ term missing?

Line 210: The NeurIPS 2020 paper "A biologically plausible neural network for slow feature analysis" may be relevant here.

Line 216: should "discreetly" be "discretely"?

**Limitations:**

As stated above, it is not clear to what extend the method of holomorphic equilibrium propagation is useful for understanding neural computation. The authors address this to some degree in the discussion, but not in detail. I think this is fine for this paper, but it is an important potential limitation.

**Strengths And Weaknesses:**

Strengths

I think the authors have presented a beautiful extension of equilibrium propagation for energy-based models with holomorphic activation functions. I found their presentation to be clear and as far as I can tell, their work is original (though I am not an expert in this area). The problem of solving the credit assignment problem using local learning rules is an important challenge in neuroscience and I believe the authors have put forth an interesting and elegant solution.

Weaknesses

Despite the elegance of the authors' solution, it is not clear if the results here are actually useful for understanding neural computation, which is in part the motivation for studying local learning rules. However, I think it is reasonable to leave such considerations for future work.

---

> ### Author Response · Authors · 2022-08-02
> **Response to reviewer EuCr**
>
> Dear reviewer EuCr,
>
> We thank you for the positive feedback and constructive comments. In the following, we address the points you raised and propose modifications to the paper in order to improve its clarity and link to biology. **All line, equation, and table numbers are given with respect to the revised version.**
>
> > For someone who is not an expert, Figure 2 was challenging for me to interpret. Perhaps the figures can be connected more directly to the equations?
>
> Thanks for making this point. We realize that Figure 2 is a bit hard to swallow. We now relabeled some of the axes, and added reference to the gradient equation (Eq. (7)) to improve the overall clarity of the figure
>
> > Line 143: it's not clear to me why $\mathrm{d} \beta = (2\pi \mathrm{i} \beta/T)\mathrm{d}t$. Is there a $e^{2\pi \mathrm{i} t/T}$ term missing?
>
> We are sorry this was not clear. The term $e^{2\pi \mathrm{i} t/T}$ is included in the $\beta$ of the right hand side, since $\beta = |\beta| e^{2\pi \mathrm{i} t/T}$. To clarify, we explicitly added the time dependence of $\beta$ in the right hand side of the equation. We also realized that the gradient formula Eq. 5 was not clearly stated. To clarify, we edited the results section. Specifically, we have formulated “Holomorphic Equilibrium Propagation” as Lemma 1, since it is really about generalizing the case of Eq. (2) to complex differentiation. We now promote the gradient formula Eq. (5) to be the actual Theorem 1 and we write the detailed proof in Appendix A.3 where the Cauchy formula is better introduced in reference to the newly added complex analysis background (Appendix A.1), and the change of variable much more detailed with two intermediary steps. We hope it is more clear now and thank you for pointing this out.
>
> > Line 210: The NeurIPS 2020 paper "A biologically plausible neural network for slow feature analysis" may be relevant here.
>
> Thanks for this pointer, this work is indeed relevant and we added a citation to our manuscript.
>
> > Line 216: should "discreetly" be "discretely"?
>
> Yes, this typo is now fixed in the revised version, thanks.
>
> > As stated above, it is not clear to what extent the method of holomorphic equilibrium propagation is useful for understanding neural computation. The authors address this to some degree in the discussion, but not in detail. I think this is fine for this paper, but it is an important potential limitation.
>
> Thanks for raising this point, we now expanded the relevant part of our discussion section to hypothesize about possible biological mechanisms. The corresponding section reads as follows:
>
> *“However, complex outputs have a long-standing tradition in computational neuroscience where they appear in complex variations of Hopfield networks [55–57], in the framework of theta neurons [58], and phasor networks [59] where they capture oscillatory neuronal dynamics with respect to a reference oscillation. It is conceivable that holomorphic EP can be interpreted within such a framework. For instance, it is straightforward to interpret complex neuronal output as oscillating activity with a defined amplitude and relative phase to some reference signal, which has to be accessible to the entire neuronal population. Such a signal could be implemented by neuromodulators such as acetylcholine which has been implicated in the regulation of the theta rhythm [60]. Within our framework, the oscillatory teaching signal then corresponds to a slow phase precession between the reference and neuronal activity. Importantly, such a mechanism implies a hierarchy of different oscillation frequencies, which are ubiquitously found in the brain, e.g., theta (4–8 Hz), gamma (30–70 Hz). Still, their precise purpose for brain function remains elusive. While establishing formal circuit-level equivalences with holomorphic EP will require future work, the principled link between oscillatory activity, learning and memory as required by the model seems promising.”*

---

> > ### Comment · Reviewer_EuCr · 2022-08-03
> > **Rebuttal response**
> >
> > Thanks for the clarifications. The authors have addressed all of my concerns. I increased my score 1 point.

---

### Official Review · Reviewer_YMzn · 2022-07-11

**Rating:** 9
**Confidence:** 4
**Soundness:** 3 good
**Presentation:** 3 good
**Contribution:** 4 excellent

**Summary:**

Equilibrium propagation is a promising framework to learn neural networks in a biologically plausible way. However, it is sensitive to noise, which makes it hard to scale to large machine learning tasks, and requires two phases, limiting its interpretability as a biologically plausible learning framework. The authors solve those two problems by introducing a new variant of equilibrium propagation that relies on complex analysis and holomorphic functions, and derive an online unbiased version of the corresponding learning rule. They show that this new algorithm is more robust to noise than traditional equilibrium propagation and scales to large vision classification tasks such as ImageNet.

**Questions:**

The suggestions below aim at improving the main weakness of the paper:

- It is hard to understand how $s_\beta^*$ is computed when $\beta$ is a complex variable. A naive interpretation would be that both the real and imaginary parts of $\partial_s F$ have to be 0. However, this does not require $s$ to be a complex variable, which seems to be the case in the paper (cf. Eq. 3 or Fig.2.a). More details on how $F$ is defined, how $s^*_\beta$ is computed, and which signal the real and imaginary parts of $s$ encode would be very valuable.

- In the paper and in the appendix, the authors assume that $\partial_s F$ is a holomorphic function. How big of an assumption is this? Do we have it for free by construction or are there some additional assumptions compared to the real version of equilibrium propagation?

- Throughout the paper, a basic understanding of complex analysis is assumed. A short background section in the appendix containing the tools (holomorphic function, Cauchy-Riemann equation, holomorphic IFT...) needed to appreciate the theoretical contributions of the paper would be useful for the reader unfamiliar with the topic.

Minor points:

Section 3 - Theoretical results
- line 122: "The proof is an extension of the one provided for the real nudging case [10]"

- In Equation 3, are the * superscript missing on $s_\beta$? the notation $s_\beta(\theta, s_\beta, \beta)$ is quite misleading, can you clarify it?

- For the Cauchy integral of equation 5 you need the implicit function $s_\beta^*$ to be defined on the path $\gamma$, which is not in the neighborhood of 0. So Theorem 1 should be more general and ensure that the IFT conditions are defined on the entire path. Can you make this point clearer, either in the main text or in the appendix?

Section 4 - Experiments
- In Table 1, the comparison between classic EP and holo EP is not fair in the noiseless case: cEP uses 200 + 1x100 steps per gradient when hEP uses 200 + 10x100. Increasing the total number of steps of cEP to 1200 (and decreasing $\beta$ to benefit from this extra number of steps) would make the comparison fairer.

- In Table 1, the cEP estimate used in the noisy regime uses 1 point when $\beta$ is zero and 10 for non-zero $\beta$. This results in a very biased negative update but unbiased positive update. One other strategy would be to use 5 points for the two phases to reduce this "bias imbalance". I would suspect this to improve the quality of the cEP estimate in the noisy regime.

- Figure 3.a) and 4.a) plot "cosine dissimilarity" when Figure 2.d) and 3.c) use "cosine similarity". Using the same measure would make navigation in the figures easier.

- Legend Figure 3.a), are $T_\text{dyn}$ and $T_\text{plas}$ fixed? If so, to which values?

- Line 236-238: refer to Zucchet et al. 2021, which theoretically analyzed this behavior?

- It is unclear to me how important is the choice of the dSiLU activation function. On one side, it seems necessary theoretically to have holomorphic dynamics. On the other, it seems to allow for a more flexible choice of $\beta$ (see Fig. 7), which could suggest that it makes convergence to equilibrium faster. I’m therefore wondering how crucial is this activation for the good empirical results displayed in the paper.

Appendix
- Equation (12): with the current brackets it is not directly that the product is 0 because $\partial_s F$ and $\partial_{\bar{s}} F$ are. Moving them under the partials would make the proof easier to read.

**Limitations:**

I found the discussion of the limitations of the paper adequate.

**Strengths And Weaknesses:**

+ Significant theoretical advances on equilibrium propagation
+ Careful empirical study of the impact of the different parameters ($N$, $\beta$) on the quality of the gradient approximation
+ Strong empirical results (robustness to noise, scale to ImageNet) while being arguably more plausible than previous equilibrium propagation versions
+ The paper is overall very well written
-  (Weakness) A detailed description of the algorithm is lacking, along with an interpretation in terms of neural dynamics

---

> ### Author Response · Authors · 2022-08-02
> **Response to reviewer YMzn Part 1**
>
> Dear reviewer YMzn,
>
> We thank you for your detailed feedback that allowed us to further improve the clarity of the paper. We address the points you raised one by one below along with proposed modifications to the manuscript. **All line, equation, and table numbers are given with respect to the revised version.**
>
> > It is hard to understand how $s^{\ast}\_{\beta}$ is computed when $\beta$ is a complex variable. A naive interpretation would be that both the real and imaginary parts of $\partial_{s} F$ have to be 0. However, this does not require $s$ to be a complex variable, which seems to be the case in the paper (cf. Eq. 3 or Fig.2.a).
>
> Thanks for the question. We realize that this wasn’t entirely clear. When $\beta$ is complex, nothing changes in the expression of $F$ or in the equations defining the dynamics of the network (detailed in Appendix B). What changes is that $F$ starts to be evaluated in the complex domain through the complex $\beta$, and the state variable $s$ takes complex values as a result of $\beta$ being complex. However, the fixed point in the complex domain is still defined as $\partial_{s} F = 0$ and obtained in the same way. To clarify this point, we edited the sentence at line 143 to make it more clear:
>
> *“Note that complex-valued teaching signals $\beta \in \gamma$ produce fixed points in the complex plane computed through the same equations as in the real case (see Appendix B).”*
>
> > More details on how $F$ is defined, how $s^{\ast}_{\beta}$ is computed [...]
>
> For complex $\beta$ the definition of $F$ does not change. However, to ensure the existence of complex derivatives, we assume that $F$ only involves holomorphic functions (in practice many common functions are holomorphic, except functions involving, e.g., max operators). To make this more clear, we modified the sentence right after Eq. (6) which now reads:
>
> *“If the activation function $\sigma$ is holomorphic, which is true in the case of sigmoid functions, the same $F$ (Eq. (6)) can be evaluated with complex $\beta$, and we can apply Eq. (5) to obtain:”*
>
> > [...] and which signal the real and imaginary parts of $s$ encode would be very valuable.
>
> This is a really interesting question. While for the first order terms in $\beta$, the real and imaginary parts encode the gradient and are linked through their time derivatives (cf. Eq. (17), higher order contributions are hard to interpret, but one of our surprising findings is that their contribution to the gradient cancels. To elucidate, we added an explanatory note to Appendix A.4 “Roles of real and imaginary parts in the learning rule” which reads as follows:
>
> *“As an interesting final note, if we define $\operatorname{Re}\_{1}(\sigma(s_{i, \beta}^{\ast}))$ and $\operatorname{Im}\_{1}(\sigma(s\_{i, \beta}^{\ast}))$, the first order contributions in $\beta$ to the real and imaginary parts of the neural activity, we find that they are the only ones to contribute to the gradient computation. We can appreciate that they are related through $\operatorname{Im}_{1}(\sigma(s\_{i, \beta}^{\ast})) = - \frac{T}{2 \pi} \frac{\mathrm{d}}{\mathrm{d}t}\operatorname{Re}\_{1}(\sigma(s\_{i, \beta}^{\ast}))$, where the time derivative is at the scale of the teaching signal.”*
>
> However, another perhaps more intuitive way to understand the complex neural activity is to consider the amplitude and phase with respect to the free fixed point rather than the real and imaginary parts independently. In the revised discussion, we further detail this interpretation.
>
> *“However, complex outputs have a long-standing tradition in computational neuroscience where they appear in complex variations of Hopfield networks [55–57], in the framework of theta neurons [58 ], and phasor networks [59] where they capture oscillatory neuronal dynamics with respect to a reference oscillation. It is conceivable that holomorphic EP can be interpreted within such a framework. For instance, it is straightforward to interpret complex neuronal output as oscillating activity with a defined amplitude and relative phase to some reference signal, which has to be accessible to the entire neuronal population. Such a signal could be implemented by neuromodulators such as acetylcholine which has been implicated in the regulation of the theta rhythm [60]. Within our framework, the oscillatory teaching signal then corresponds to a slow phase precession between the reference and neuronal activity. Importantly, such a mechanism implies a hierarchy of different oscillation frequencies, which are ubiquitously found in the brain, e.g., theta (4–8 Hz), gamma (30–70 Hz). Still, their precise purpose for brain function remains elusive. While establishing formal circuit-level equivalences with holomorphic EP will require future work, the principled link between oscillatory activity, learning and memory as required by the model seems promising.”*

---

> ### Author Response · Authors · 2022-08-02
> **Response to reviewer YMzn Part 2**
>
> > In the paper and in the appendix, the authors assume that $\partial_{s} F$ is a holomorphic function. How big of an assumption is this? Do we have it for free by construction or are there some additional assumptions compared to the real version of equilibrium propagation?
>
> The assumption of $F$ being holomorphic in its variables only precludes the use of activation functions using max operators such as ReLU or hard sigmoids. Other usual functions like exponential, cosine, sine, sigmoids etc. are all holomorphic. In this sense, the requirements of $F$ being differentiable with respect to complex variables are not very strong. However, the gains of this relatively weak assumption are high. In particular, what follows is that $\partial_{s} F$ is holomorphic and differentiable to any order. These facts are now stated in Appendix A.1.
>
> > Throughout the paper, a basic understanding of complex analysis is assumed. A short background section in the appendix containing the tools (holomorphic function, Cauchy-Riemann equation, holomorphic IFT...) needed to appreciate the theoretical contributions of the paper would be useful for the reader unfamiliar with the topic.
>
> Thanks for this suggestion, we added a background section, as suggested, in the new Appendix A.1 with the minimal toolkit of complex analysis to fully appreciate the theoretical contributions of our work.
>
> > line 122: "The proof is an extension of the one provided for the real nudging case [10]"
>
> We were not sure what the reviewer meant with this suggestion, but reworded the corresponding sentence and moved the position of the citation. It now reads:
>
> *“The proof is an extension of the one provided by [10] for the real nudging case.”*
>
> > In Equation 3, are the $^{\ast}$ superscript missing on $s_{\beta}$?
>
> Thanks for noticing this, we rectified this discrepancy in the manuscript. Since the notion with the $^{\ast}$ superscript is not formally required, we removed it throughout the manuscript to keep the notation consistent.
>
> > The notation $s_{\beta}(\theta, s_{\beta}, \beta)$ is quite misleading, can you clarify it?
>
> You are right. This was a copy and paste error. Thanks for pointing it out. We now removed the $s_{\beta}$ argument.
>
> > For the Cauchy integral of equation 5 you need the implicit function $s^{\ast}_{\beta}$ to be defined on the path $\gamma$, which is not in the neighborhood of 0. So Theorem 1 should be more general and ensure that the IFT conditions are defined on the entire path. Can you make this point clearer, either in the main text or in the appendix?
>
> We thank the reviewer for this  comment. We realize that we did not clearly state  Theorem 1 and the gradient formula Eq. 5 with respect to each other. To clarify, we edited the results section. Specifically, we have formulated “Holomorphic Equilibrium Propagation” as Lemma 1, since it is really about generalizing the case of Eq. (2) to  complex differentiation. And we now promote the gradient formula Eq. (5) to be the actual Theorem 1 and we restate our assumptions about the path lying in the domain of $s^{\ast}_{\beta}$ more clearly. We think this substantially improved the clarity of the argument.

---

> ### Author Response · Authors · 2022-08-02
> **Response to reviewer YMzn Part 3**
>
> > In Table 1, the cEP estimate used in the noisy regime uses 1 point when beta is zero and 10 for non-zero beta. This results in a very biased negative update but unbiased positive update. One other strategy would be to use 5 points for the two phases to reduce this "bias imbalance". I would suspect this to improve the quality of the cEP estimate in the noisy regime.
>
> Thanks for pointing this out, we re-ran the classic EP simulations with averaging also over the free equilibrium as suggested. This makes classic EP perform slightly worse, since before we used the unperturbed free equilibrium which can be seen as obtained as an average over infinite samples. We agree that this comparison is now more fair and we updated all results accordingly.
>
> > In Table 1, the comparison between classic EP and holo EP is not fair in the noiseless case: cEP uses 200 + 1x100 steps per gradient when hEP uses 200 + 10x100. Increasing the total number of steps of cEP to 1200 (and decreasing beta to benefit from this extra number of steps) would make the comparison fairer.
>
> Thanks for spotting this, while running the suggested comparison, we realized that we had made a copy-paste mistake while reporting the hyperparameters of holomorphic EP in Table 3. Holomorphic EP actually used 200 + 10x**50** = 700 time steps instead of 200 + 10x100. This can be seen, for instance, in  `code/results/table1/no_noise/hep/*/hyperparameters.json` with the T2 argument equal to 50 and not 100. Our apologies for this oversight. To correct this problem, we reran the classic EP simulations without noise now using 350 time steps in the free phase and 350 steps in the nudged phase, which amounts to 700 time steps total. We also added the corresponding results in Table 1, and we report error values for both $\beta = 0.4$ and 0.1. While the performance of classic EP indeed increased compared to the noise-free $\beta = 0.4$ case, it was unable to learn in the presence of noise even with the suggested fair averaging. To reflect this new findings, we also updated Fig 3c with the fair averaging strategy for classic EP. Finally, we amended Table 3 with the correct hyperparameters.
>
> > Figure 3.a) and 4.a) plot "cosine dissimilarity" when Figure 2.d) and 3.c) use "cosine similarity". Using the same measure would make navigation in the figures easier.
>
> Fair point, thanks for raising it. We now revised the relevant figures and consistently use cosine similarity throughout the paper.
>
> > Legend Figure 3.a), are $T_{\text{dyn}}$ and $T_{\text{plas}}$ fixed? If so, to which values?
>
> In Fig. 3.a), $T_{\text{dyn}}$ is about 400/10=40 time steps as measured by the time for which the fixed point between two consecutive complex $\beta$ is reached (Fig 3.b)). Fig 3.a) also shows how $T_{\text{plas}}$ could be chosen as $10T_{\text{osc}}$ so that the weight update optimally follows the gradient. We modified the legend of Fig 3.a) to make these points explicit.
>
> > Line 236-238: refer to Zucchet et al. 2021, which theoretically analyzed this behavior?
>
> Thanks for pointing this out. We initially just meant that in the presence of noise, a non-zero $\beta$ becomes optimal for classic EP as shown in Fig 3.c). We realize now that this statement is indeed related to the theory of Zucchet et al. 2021 showing that non-zero $\beta$ is optimal when the fixed point is not reached perfectly. We added the suggested reference to make it more explicit.
>
> > It is unclear to me how important is the choice of the dSiLU activation function. On one side, it seems necessary theoretically to have holomorphic dynamics. On the other hand, it seems to allow for a more flexible choice of $\beta$ (see Fig. 7), which could suggest that it makes convergence to equilibrium faster. I’m therefore wondering how crucial is this activation for the good empirical results displayed in the paper.
>
> Thanks for raising this point. Most existing works on EP in CNNs [r1,r2,r3] rely on a hard sigmoid. To stay as close as possible to this architecture while also ensuring holomorphicity, we tried sigmoid and dSiLU and found empirically that dSiLU works better than sigmoid. One reason could be that dSiLU ‘saturates’ units less than sigmoids, which may reduce the risk of vanishing gradients, but we did not notice faster convergence to equilibrium.
>
> [r1] Ernoult et al. NeurIPS 2019
>
> [r2] Laborieux et al. Frontiers in Neuroscience 2021
>
> [r3] Luczak et al, Nature Machine Intelligence 2022
>
> > Equation (12): with the current brackets it is not directly that the product is 0 because $\partial_{s} F$ and $\partial_{\bar{s}} F$ are. Moving them under the partials would make the proof easier to read.

---

> > ### Comment · Reviewer_YMzn · 2022-08-08
> > **Thank you for your replies**
> >
> > I would like to thank the authors for their replies which made me understand the paper better, and which made an already strong paper even stronger.
> > I am now ready to vote for very strong acceptance for this paper, which I believe is a significant advance on an important research topic.

---

### Official Review · Reviewer_8a15 · 2022-07-18

**Rating:** 8
**Confidence:** 2
**Soundness:** 3 good
**Presentation:** 3 good
**Contribution:** 3 good

**Summary:**

By restricting (can also be viewed as expanding) to holomorphic neural networks, the authors are able to use a contour integral in the complex plane for finite $\beta$ to evaluate a differential used in equilibrium propagation (EP) that is only valid as an approximation of the backprop gradient in the limit $\beta \to 0$. The authors interpret this use of the contour integral as a time integral of an oscillating teaching signal. The authors show that this holomorphic EP surpasses EP in performance especially in the presence of noise.


**Questions:**

Perhaps the authors can speculate further in the discussion how such a scheme may be implemented bio-plausibly.

[Update: since the authors have included this, I have enhanced my rating from 7 to 8.]

Minor corrections:
Figure 2 legend for panel d "circular path in panel d)" -> "circular path in panel b".
inconsistency in lines 85-86: $l(y,\theta,s)$ on line 85 vs $l(\theta,s,y)$ on line 86, also line 91.


**Limitations:**

Yes

**Strengths And Weaknesses:**

The authors incorporate a cool idea to convert a differential to a contour integral in the complex plane and then interpret this integral as a time integral of an oscillating teaching signal. They also show that allowing for finite beta enables robustness to noise, unlike EP which is only vaild in the lim $\beta \to 0$. However, the authors do not provide any pointers as to how this may be implemented in biology except mentioning that there may be connections to theta neurons and phasor networks. Still this is an interesting idea that possibly deserves to be disseminated to stimulate further progress in the search for bio-plausible back-prop.
Disclaimer: I have not verified any proofs / derivations.

---

> ### Author Response · Authors · 2022-08-02
> **Response to reviewer 8a15**
>
> Dear reviewer 8a15,
>
> Thanks for the positive assessment of our work, the comments, and questions concerning possible biological implementations.
>
> > Perhaps the authors can speculate further in the discussion how such a scheme may be implemented bio-plausibly.
>
> While establishing formal circuit-level equivalences with hEP will require future work, we see natural links between neural oscillations which are pervasive in the brain and the complex valued activity in our model. For instance, particularly plastic brain areas such as the hippocampus are known to exhibit a diversity of oscillation frequencies. It is conceivable that neurons rely on the phase of a fast oscillation such as the gamma rhythm (30-70Hz) to encode complex phase as defined in the framework of hEP, whereas the teaching signal introduced in our article would require a periodic modulation of its phase with a slower frequency, e.g., the theta rhythm (4-8Hz). To address this point the article, we now updated our discussion by adding the following sentences:
>
> *“However, complex outputs have a long-standing tradition in computational neuroscience where they appear in complex variations of Hopfield networks [55–57], in the framework of theta neurons [58], and phasor networks [59] where they capture oscillatory neuronal dynamics with respect to a reference oscillation. It is conceivable that holomorphic EP can be interpreted within such a framework. For instance, it is straightforward to interpret complex neuronal output as oscillating activity with a defined amplitude and relative phase to some reference signal, which has to be accessible to the entire neuronal population. Such a signal could be implemented by neuromodulators such as acetylcholine which has been implicated in the regulation of the theta rhythm [60]. Within our framework, the oscillatory teaching signal then corresponds to a slow phase precession between the reference and neuronal activity. Importantly, such a mechanism implies a hierarchy of different oscillation frequencies, which are ubiquitously found in the brain, e.g., theta (4–8 Hz), gamma (30–70 Hz). Still, their precise purpose for brain function remains elusive. While establishing formal circuit-level equivalences with holomorphic EP will require future work, the principled link between oscillatory activity, learning and memory as required by the model seems promising.”*
>
> > Minor corrections: Figure 2 legend for panel d "circular path in panel d)" -> "circular path in panel b". inconsistency in lines 85-86: l(y, theta, s) on line 85 vs l(theta, s y) on line 86, also line 91.
>
> Thanks for pointing these typos out. These have been fixed in the revised version of the paper.

---

> > ### Comment · Reviewer_8a15 · 2022-08-09
> > **response**
> >
> > Thanks for enhancing the discussion with pointers towards a bioplausible implementation.
> > I am increasing my rating from 7 to 8.

---

### Author Response · Authors · 2022-08-09
**Thanks to all the reviewers**

Dear reviewers,

We would like to thank you again for the time you spent in reviewing the manuscript. We truly appreciated your valuable comments and suggestions, which helped us improve the clarity of the paper.

---

### Meta-Review · Area_Chair_KwhH · 2022-08-25

**Recommendation:** Accept
**Confidence:** Certain

**Metareview:**

Equilibrium propagation is a biologically plausible form of backpropagation based learning where the true gradient of an energy based model is computed for infinitesimal perturbations. In this innovative work, authors extend EP using complex analysis that links contour integrals for finite perturbations with the oscillatory dynamics in time. This not only allows better gradient estimates, but also applications to related theories of learning in neuroscience as well as neuromorphic engineering. It represents a significant advance that opens new doors.


**Award:**

No

---

### Decision · Program_Chairs · 2022-09-14

Accept